# Characterization of Peripheral Blood TCR in Patients with Type 1 Diabetes Mellitus by BD Rhapsody^TM^ VDJ CDR3 Assay

**DOI:** 10.3390/cells11101623

**Published:** 2022-05-12

**Authors:** Takuro Okamura, Masahide Hamaguchi, Hiroyuki Tominaga, Noriyuki Kitagawa, Yoshitaka Hashimoto, Saori Majima, Takafumi Senmaru, Hiroshi Okada, Emi Ushigome, Naoko Nakanishi, Shigeyuki Shichino, Michiaki Fukui

**Affiliations:** 1Department of Endocrinology and Metabolism, Graduate School of Medical Science, Kyoto Prefectural University of Medicine, Kyoto 602-8566, Japan; d04sm012@koto.kpu-m.ac.jp (T.O.); mhama@koto.kpu-m.ac.jp (M.H.); htommy@koto.kpu-m.ac.jp (H.T.); nori-kgw@koto.kpu-m.ac.jp (N.K.); y-hashi@koto.kpu-m.ac.jp (Y.H.); saori-m@koto.kpu-m.ac.jp (S.M.); semmarut@koto.kpu-m.ac.jp (T.S.); conti@koto.kpu-m.ac.jp (H.O.); emis@koto.kpu-m.ac.jp (E.U.); naoko-n@koto.kpu-m.ac.jp (N.N.); 2Division of Molecular Regulation of Inflammatory and Immune Diseases, Research Institute of Biomedical Sciences, Tokyo University of Science, Chiba 278-0022, Japan; s_shichino@rs.tus.ac.jp

**Keywords:** single-cell RNA sequencing, type 1 diabetes, genetic research, genetic screening, T-cell receptor

## Abstract

The sequence of complementarity-determining region 3 of the T-cell receptor (TCR) varies widely due to the insertion of random bases during V-(D)-J recombination. In this study, we used single-cell VDJ sequencing using the latest technology, BD Rhapsody, to identify the TCR sequences of autoreactive T-cells characteristic of Japanese type 1 diabetes mellitus (T1DM) and to clarify the pairing of TCR of peripheral blood mononuclear cells from four patients with T1DM at the single-cell level. The expression levels of the TCR alpha variable (TRAV) 17 and TRAV21 in T1DM patients were higher than those in healthy Japanese subjects. Furthermore, the Shannon index of CD8^+^ T cells and FOXP3^+^ cells in T1DM patients was lower than that of healthy subjects. The gene expression of PRF1, GZMH, ITGB2, NKG7, CTSW, and CST7 was increased, while the expression of CD4, CD7, CD5, HLA-A, CD27, and IL-32 was decreased in the CD8^+^ T cells of T1DM patients. The upregulated gene expression was IL4R and TNFRSF4 in FOXP3^+^ cells of T1DM patients. Overall, these findings demonstrate that TCR diversity and gene expression of CD8^+^ and FOXP3^+^ cells are different in patients with T1DM and healthy subjects.

## 1. Introduction

T-cell receptors (TCRs) contain a stationary region (C region) and a variable region (V region), and the structure of the variable region acquires remarkable diversity by reconfiguration of the TCR and immunoglobulin genes. In the case of TCRβ, TCRδ, and immunoglobulin H chains (heavy chains), three complementarity-determining regions called complementarity-determining region (CDR) 1, CDR2, and CDR3 are formed by the recombination of the V, D, and J gene segments. In both TCR and B-cell receptors (BCRs), CDR1 and CDR2 are encoded in the V segment, and the same gene sequences as in the germline are used. Conversely, the CDR3 sequence can have significant diversity, even when the same V, D, and J segments are used, due to the insertion of random bases (N sequences) during V-(D)-J reconstruction.

It is widely known that abnormalities in B- and T-cell repertoires are involved in the development and progression of both autoimmune and allergic diseases, but the details have not been fully elucidated. Recently, comprehensive immunosequencing has been used to analyze the pathogenesis of type 1 diabetes mellitus (T1DM), rheumatoid arthritis, systemic lupus erythematosus, and multiple sclerosis [1]. One research group conducted a comprehensive immunosequencing analysis of TCRs and BCRs using lymphocytes isolated from the peripancreatic lymph nodes and spleens of T1DM donors registered with the Network for Pancreatic Organ Donors with Diabetes (nPOD). They reported the isolated lymphocytes to have a high frequency of T-cells expressing the TCR beta chain, similar to GAD14.3, a known glutamic acid decarboxylase 65 (GAD65)-reactive T-cell clone, in the lymphoid tissues of patients with T1DM [2].

The appearance of effector T-cells that respond to autologous islet antigens is considered a critical factor in T1DM. It is assumed that effector T-cells with a unique TCR that reacts with autologous islet antigens are monoclonal and activated [3]. Moreover, cytotoxic CD8^+^ T cells are thought to play a major role in the destruction of beta cells during the development of T1DM [4,5], and TCR variability of regulatory T-cells (Tregs) has been proposed to be beneficial for the maintenance of self-tolerance [6]. Therefore, it is important to clarify the variability of TCRs in CD8^+^ T cells and Tregs in order to elucidate the pathogenesis of T1DM. The antigen-binding sites for TCRs are determined by genetic rearrangements and have acquired diversity in the order of 10^10^ combinations. In the past, it was difficult to determine the full extent of these vast repertoires of antigen receptors; however, with the dramatic development of next-generation sequencing (NGS) technology, it is now possible to identify the gene sequences of TCRs expressed in a desired cell population at the individual clone level. Currently, such comprehensive immunosequencing technology is being applied to the in vivo monitoring of immune responses and drug discovery of antibody drugs, vaccines, and cellular drugs and is expected to bring about significant innovations in various medical fields in the future.

In this study, we will reveal the TCR sequences of characteristic autoreactive T-cells in Japanese patients with T1DM by single-cell VDJ sequencing.

## 2. Materials and Methods

### 2.1. Study Design and Participants

The KAMOGAWA-DM cohort study is an ongoing prospective cohort study that was approved by the Ethics Committee of the Kyoto Prefectural University of Medicine in 2013 (Kyoto, Japan, RBMR-E-466) [7]. Informed consent was obtained from all patients involved in the KAMOGAWA-DM cohort study. We randomly selected 4 patients with T1DM who visited our diabetes outpatient clinic from April to May 2021. In addition, PBMCs were collected on the day of the visit, and the experiment was conducted on the same day using fresh specimens without cryopreservation. In addition, none of those four patients had any apparent infection during the study period. T1DM was diagnosed based on the criteria of the American Diabetes Association [8]. According to the recommendation of the Committee of Experts of the American Diabetes Association, T1DM was divided into type 1A diabetes (i.e., immune-mediated), type 1 B (i.e., other forms of diabetes with severe insulin deficiency but without proof of autoimmune etiology, also known as idiopathic) [9], and slowly progressive insulin-dependent diabetes mellitus (SPIDDM) at all participating institutions in this study [10,11,12,13].

Furthermore, we used the scRNA-seq data generated from a healthy subject in a previous study from the NCBI’s Gene Expression Omnibus under accession number GSE150060 [14].

### 2.2. Data Collection

Information regarding patients’ background demographics (i.e., age, sex, disease duration, and smoking habits) was gathered from their medical records. Blood pressure was measured in an outpatient clinic. After an overnight fast, venous blood samples were collected to measure fasting plasma levels of glucose, C-peptide, triglycerides, total cholesterol, high-density lipoprotein cholesterol, low-density lipoprotein cholesterol, creatinine, and uric acid. The hemoglobin A1c level was determined by high-performance liquid chromatography and presented herein using the National Glycohemoglobin Standardization Program unit.

### 2.3. BD Rhapsody Single Cell Analysis System

Heparin was added to the syringe when peripheral blood was collected. Peripheral blood mononuclear cells (PBMCs) were isolated by density gradient centrifugation, counted, and resuspended in 650 μL of cold sample buffer for loading on a BD Rhapsody cartridge. Targeted scRNA-seq with TCR analysis was performed using the BD Rhapsody Express system (BD Biosciences, Piscataway, NJ, USA). Cell capture and library preparation were performed using the BD Rhapsody Targeted mRNA and AbSeq Reagent Kit (BD Biosciences), according to the manufacturer’s instructions. Briefly, 1 × 10^4^ cells from each fresh blood sample were captured in a microwell plate with beads. This was followed by cell lysis, bead retrieval, cDNA synthesis, template switching, Klenow extension, and library preparation (a targeted gene library using a human T-cell expression panel and a TCR gene library) following the BD Rhapsody VDJ CDR3 protocol. The final pooled libraries were spiked with 20% PhiX control DNA to increase the sequence complexity and subsequently sequenced (75 bp × 225 bp paired-end) on a HiSeq X Ten sequencer (Illumina, San Diego, CA, USA). In this study, HLA typing was not tested because the number of cells in each sample was not sufficient.

### 2.4. Data Analyses

The FASTQ files obtained from the sequences were processed using the BD Rhapsody Targeted Analysis Pipeline with V(D)J processing incorporated (BD Biosciences) in the Seven Bridges Platform (https://www.sevenbridges.com/d, accecessed on 20 October 2021). First, low-quality read pairs were removed based on read length, average base quality score, and highest single-base frequency. High-quality R1 reads were analyzed to identify cell labels and unique molecular identifier (UMI) sequences. The high-quality R2 reads were aligned with the reference panel sequences (Human T cell Expression panel) and TCR gene segments from the international ImMunoGeneTics information system^®^ (IMGT.org) using the program Bowtie2. IGBlast was utilized for CDR3 determination. Reads with identical cell labels, identical UMI sequences, and identical genes were folded into a single molecule. The obtained counts were subjected to error correction algorithms (recursive substitution error correction (RSEC) and distribution-based error correction (DBEC)) developed by BD Biosciences. The DBEC-adjusted number of molecule data obtained from the Rhapsody pipeline was imported into SeqGeq version 1.6.0, and quality control was then performed to gate out cells that were significantly smaller and with low numbers of expressed genes (dead cells). Subsequently, dimensional reduction and unbiased clustering in SeqGeq were performed using the Seurat plug-in. Briefly, Seurat was set up to include all genes used, and the QC function, log normalization, and UMAP (uniform manifold approximation and projection) were used for dimensionality reduction. These plug-ins output data, including UMAP, lists of upregulated and downregulated genes, and annotation information, using the PBMC gene model. Further clustering analysis was completed with manual curation. Integration of the cluster information and TCR CDR3 information in each cell was performed using the VDJExploler plug-in of SeqGeq. As a parameter for the structural diversity of the genes, the Shannon index H′ was calculated [15].
$$H^{\prime }=-\overset{S}{\underset{i=1}{\sum }}p_{i}lnp_{i}$$

*S*: Number of genes observed in the sample

*p_i_*: Ratio of genes *i* to the total sample

### 2.5. Detection of CD8^+^ Cells and FOXP3 Expression in Tregs Using Whole Transcriptome scRNA-seq Data

FOXP3 expression was assessed in two publicly available genomic datasets, combining 3 mRNA and surface protein expression datasets. Then, 10k PBMC datasets were generated using v3 chemistry (7865 cells passing QC, average reads per cell of mRNA library 10k PBMC dataset generated using v3 chemistry (7865 cells passing QC, average reads per cell for mRNA libraries: 35,433) and the 5k PBMC dataset generated using NextGEM chemistry (5527 cells passing QC, average reads per cell for mRNA libraries: 30,853). See https://support.10xgenomics.com/single-cell-gene-expression/datasets/, accessed on 20 October 2021). CD8^+^ or FOXP3^+^ T cells were defined as cells expressing one or more copies of CD8 or FOXP3 (UMI).

### 2.6. TCR CDR3 Motif Identification

All TCR CDR3 amino acid sequences from the current study were aligned using the MEME suite (https://meme-suite.org/meme/tools/meme) [16].

## 3. Results

### 3.1. Single-Cell mRNA Immunophenotyping Identifies Distinct Trajectories of T-Cell Differentiation in Blood

Sample 1 (S1) was from a male patient with type 1A T1DM, Sample 2 (S2) was from a female patient with type 1B T1DM, Sample 3 (S3) was from a patient with type 1B T1DM, and Sample 4 (S4) was obtained from a patient with SPIDDM (Table 1). Using immunosequencing, we attempted to comprehensively analyze the TCRs expressed by T-cells in the peripheral blood of patients.

First, to determine the usage rate of TCR variable (TRV) and TCR joining (TRJ) genes, we counted the number of copies (reads) of USRs containing each TRV and TRJ. For the TCR alpha (TRA) repertoire, eight pseudogenes (AV8-5, AV11, AV15, AV28, AV31, AV32, AV33, and AV37) were not expressed in each patient. AV8-7 was classified as an ORF (defined by IMGT based on the sequence of splicing sites, recombination signals, and regulatory elements), which was also not expressed. Moreover, AV7, AV8-6-1, AV8-7, AV9-1, AV14-1, AV18, and AV46 were not expressed in any patient. The majority of TRA in S1, S2, and S3 was AV9-2, AV12-1, AV12-3, AV13-1, and AV17, whereas in S4, AV12-1, AV12-3, and AV13-1 were common to S1, S2, and S3; however, AV-2 and AV13-2 had specifically high rates (Figure 1).

None of the patients expressed any of the three pseudogenes AJ51, AJ55, and AJ60, while AJ1, AJ2, AJ14, AJ19, AJ25, AJ59, and AJ61 were not expressed in any of the patients. In S1, S2, and S3, the expression of AJ9, AJ20, AJ29, AJ34, and AJ49 had high rates, whereas in S4, AJ9 was common to S1, S2, and S3; however, AJ11, AJ40, AJ42, and AJ53 had specifically high rates (Figure 2).

As for TCR beta (TRB) genes, out of the 5 pseudogenes (BV1, BV3-2, BV12-1, BV12-2, and BV21-1), BV1, BV3-2, BV12-2, and BV21-1 were expressed. Of the six ORF genes (BV5-3, BV5-7, BV6-7, BV7-1, BV17, and BV23-1), only BV23-1 was expressed (Figure 3).

Of BJ, in each patient, BJ1-1, BJ2-1, BJ2-2, BJ2-3, and BJ2-7 were in the majority (Figure 4).

### 3.2. Recombination of TRAV and TRAJ

The genetic recombination of 41 TRAVs and 50 TRAJs (excluding pseudogenes, ORFs, and low-expressed genes) resulted in a total of 2050 AV-AJ recombinants. Notably, the AV1-1-AV6 gene did not preferentially combine with the AJ50-AJ58 gene, and similarly, little recombination was observed between the AV35-AV41 gene and AJ3-AJ16. For TRB, 650 genetic recombinations occurred in 50 BV genes (excluding 11 pseudogenes and 5 ORFs) and 13 BJ genes (excluding pseudogenes). There were no restrictions on the combination of TRBV and TRBJ, as observed in the TRA. The Shannon index H’ was used as a diversity index to evaluate the diversity of TRA and TRB. Shannon-index H’ of TRA and TRB in S1 was 10.80 and 10.83, that in S2 was 11.62 and 11.69, that in S3 was 10.26 and 10.57, and that in S4 was 11.37 and 11.25, respectively (Table 2).

No significantly high repertoires were identified in S1, but AV12-1/AJ45, AV12-3/AJ49, AV12-3/AJ54, and AV9-2/AJ57 in S2, AV1-2/AJ33, AV12-2/AJ21, AV21/AJ20, and AV12-3/AJ40 in S3, and AV12-1/AJ11, AV2/AJ40, and AV13-2/AJ56 in S4 were expressed at a significantly high rate (Figure 5).

BV24-1/BJ2-1 in S1; BV28/BJ2-1, BV5-1/BJ2-7, BV29-1/BJ2-7, and BV29-1/BJ2-1 in S2; BV7-8/BJ2-1, BV7-2/BJ2-7, BV29-1/BJ2-1, BV29-1/BJ2-1, BV29-1/BJ2-3 in S3; and BV20-1/BJ2-7, BV5-1/BJ2-1, BV7-7/BJ2-2, and BV9/BJ2-2 in S4 were expressed at a significantly high rate, respectively (Figure 6). Taken together, there was no common pattern among each of the four samples and healthy controls.

### 3.3. TCR Clonotypes of CD8^+^ T Cells and FOXP3^+^ T Cells

Next, we investigated the genetic recombination of TRAVs and TRAJs and that of TRBVs and TRBJs in CD8^+^ T cells and FOXP3^+^ T cells, and we have shown the top ten TCR clonotypes in Table 3 and Table 4. The most frequently observed CDR3 of TRA and TRB in CD8^+^ T cells in S1 were AGAISNNDMR and ASSVVGSGTDEQF; those of S2 were VVRARPPLPWSGGGADGLT and ASTPPSSPGYEQY, those of S3 were AFSGGYQKVT and ASSLAGEGSGTGELF, and those of S4 were VVSAFFSGGSYIPT and ASSSSRDRGNYEQY (Table 3).

In the FOXP3^+^ T cells of S1 were AMRFKSGYNKLI and ASSPPTSGASYEQY; those of S2 were ALSSNDYKLS and ASTLDGPGSPLH, those of S3 were GFSSGSARQLT and ASSFGRYEQY, and those of S4 were AAGRGNNRLA and ASSRTGGGYGYT (Table 4).

### 3.4. T-Cells in Blood of Patients with T1DM Have Phenotypic Hallmarks

Next, we performed an unbiased analysis of gene expression using Seurat and identified T-cell clusters in four T1DM patients (Appendix A). Heatmaps of gene expression in each cluster are shown, with 10 clusters in S1, 11 clusters in S2, 19 clusters in S3, and 11 clusters in S4 (Figure 7).

The rank of the CDR3 repertoires is shown in Appendix A. In addition, we performed clustering analyses using Seurat in CD8^+^ T cells or FOXP3^+^ T cells (Appendix A).

CDR3 motifs and clustering were shown in Appendix A. In addition, the motif-based sequence analysis tool, Multiple Em for Motif Elicitation (MEME), was then used to identify the consensus amino acids for the grouped CDR3 sequences. The top five CDR3 motifs of TRA and TRB are shown in Appendix A. We further examined which cluster CD8^+^ and Foxp3^+^ T cells with the top five CDR3 motif sequences detected belonged to (Appendix A).

For further analysis, we investigated the upregulated gene expression in CD8^+^ or FOXP3^+^ T cells of four samples compared to that in healthy subjects (S0). The legalism of all gene expressions compared to S0 is shown in the heatmaps (Appendix A). The upregulated gene expressions in CD8^+^ T cells of S1 were PRF1, GZMH, ITGB2, NKG7, and SELPLG; those of S2 were GZMH, ITGB2, PRF1, NKG7, and GNLY; those of S3 were ITGB2, GZMH, SELL, PRF1, and SELPLG; and those of S4 were GZMH, CTSW, PRF1, ITGB2, and CX3CR1, respectively. Conversely, the downregulated gene expressions in CD8^+^ T cells of S1 and S2 were CD7, CD4, CD5, CD27, and CD69, and those of S3 and S4 were CD7, CD4, CD5, CD27, and TRAC, respectively. The upregulated gene expressions in FOXP3^+^ T cells of S1 were HLA-DMA, IL4R, LIF, TNFRSF4, IL31, HLA-DMA, IL4R, TRIB2, LIF, and TNFRSF4; those of S3 were HLA-DMA, IL4R, TNFRSF4, LIF, and PRDM1, and those of S4 were HLA-DMA, IL4R, TRIB2, LIF, and TNFRSF4, respectively. The downregulated gene expressions in FOXP3^+^ cells of S1 were CD4, CD7, CD5, HLA-A, and IL32, while those of S2, S3, and S4 were CD4, CD7, CD5, HLA-A, and CD27 (Figure 8) and clustering was shown in Appendix A.

## 4. Discussion

We used adapter ligation-mediated PCR, a bias-free PCR technique, for TCR repertoire analysis using NGS. This method uses a single set of primers to avoid PCR bias due to primer competition. This method is, therefore, suitable for accurately estimating the abundance of each TCR gene in a wide variety of samples. In the present study, we comprehensively investigated the TRA and TRB repertoires from four patients with T1DM at the clonal level and evaluated a large amount of sequence data. This is the first study to reveal the TRA and TRB repertoires of patients with T1DM using BD Rapsody. Moreover, this integrated analysis makes it easy to detect the preferential use of specific TRVs and TRJs, which may be useful in studying immune responses by antigen-specific T-cells.

There are several single-cell sequencing platforms that have been widely used around the world in recent years. BD Rhapsody, which was used in this study, uses microwells and magnetic beads to isolate cells and perform bead-based 3′ RNA-Seq, thus realizing highly accurate sample preparation with low doublets and cross-contamination rates. Compared to the widely used 10× Genomics, the library cost is lower, and cell viability demand does not need to be as high as 50%, cell loading can be up to 40,000 cells, and the frequency of doublets is lower [17]. Therefore, information on a large number of cells can be obtained.

The expression levels of TRAV17, a variable gene known to be enriched in a population of CD1b-restricted T-cells [18,19], and TRAV21 in patients with T1DM were higher than those in healthy Japanese subjects investigated in a previous study [20]. Conversely, the expression levels of TRAJ in patients with T1DM were not different from those of healthy subjects. The majority of TRBV in the healthy Japanese subjects was TRBV 29-1, whereas, in the patients with T1DM, there was no clear majority compared to the healthy subjects. The Shannon indices of TRA and TRB in healthy subjects in the previous report were both approximately 7, which were clearly smaller than those of the patients with T1DM observed in this study. Moreover, we surveyed the diversity index of CD8^+^ and FOXP3^+^ T cells, and the Shannon index of the cells in the patients with T1DM was lower than those of healthy subjects [21,22]. TCR variability of Tregs has been proposed to be beneficial in the maintenance of self-tolerance [6]; therefore, the findings in this study indicate that the reduced TCR diversity in Tregs of patients with T1DM in this study may indicate reduced immune tolerance in patients with T1DM.

Upregulated genes in CD8^+^ T cells of T1DM patients included cytotoxicity-associated genes, such as PRF1, GZMH, ITGB2, NKG7, CTSW, and CST7, whereas the expression of CD4, CD7, CD5, HLA-A, CD27, and IL-32 was downregulated. Cytotoxic CD8^+^ T cells are considered to be the primary mediators of β-cell injury, based on the predominance of CD8 T-cells in pancreatic islet infiltration [23,24], as well as numerous studies using animal models of T1DM caused by β-cell injury by CD8 T-cells [25,26]. In addition, several human studies have reported an expanded pool of memory T-cells in the peripheral blood of patients with type 1 diabetes [27] and resistance of effector T-cells to Treg suppression, and our results are consistent with these previous reports [28].

Upregulated genes in FOXP3^+^ T cells in T1DM patients included IL4R. Interleukin 4 (IL-4) has been reported to be involved in several signaling pathways in the regulation of Treg cell development and function [29,30,31,32,33]. IL-4 is a cytokine that defines the type 2 immune response, while IL-4 receptor alpha (IL-4 Rα) suppresses Treg cell function during type 2 disease [34,35]. Recent reports have shown that enhanced IL-4Rα signaling by gain-of-function mutations [32,35] or chronic type 2 inflammation [36] drastically reduces the number of Foxp3^+^ Treg cells, impairs the suppressive function of Treg cells, and promotes their reprogramming to T helper 2 (Th2)-like or T helper 17 (Th17)-like cells. This receptor is further thought to play a role in suppressing Treg cell function. Although no difference in the frequency of Tregs in peripheral blood isolated from T1D patients has been reported, defects in the phenotype and suppressive capacity of Tregs have been reported [37,38,39,40,41]. In this study, we used single-cell sequencing for the first time in the world to reveal that IL4R expression is upregulated in the Tregs of patients with type 1 diabetes. As previously reported, this result suggests that the function of Tregs in T1DM is impaired. In addition, TNFRSF4 was upregulated in FOXP3^+^ T cells in patients with T1DM. TNFRSF4 is one of the most highly expressed genes in Tregs [42,43]. In addition, TNFRSF4 is one of the most highly expressed genes in tumor-invasive Tregs compared to those in healthy tissues [44,45]. While there have been no reports on the expression of TNFRSF4 in relation to Tregs in patients with type 1 diabetes, this has been reported to be significantly increased in patients with relapsed acute myeloid leukemia compared to healthy donors [46]. TNFRSF4 mediates TRAF2 and TRAF5 to activate the NF-κB pathway of TNFRSF4, and the PI3K/PKB and NFAT pathways have also been identified [47,48]. The most important function of TNFRSF4 is to promote T-cell division, proliferation, survival, and cytokine production by activating the aforementioned pathways. In this study, we found that the expression of TNFRSF4 was upregulated in T1DM regulatory T-cells, suggesting that these cells may have some immune abnormalities.

The strength of this study is that it is the first to use the BD Rhapsody system, a state-of-the-art technology for single-cell sequencing of TCR repatriation and gene expression in peripheral blood T-cells from patients with T1DM. However, this study has several limitations. First, we analyzed CD8- and FOXP3^+^ T cells as cells with more than one read, and Tregs as FOXP3^+^ T cells could also be activated T-cells and not Tregs; it would have been more accurate if we sorted each positive cell by cell sorter and performed the same analysis. Second, peripheral blood of healthy subjects was used as a control, but this data was obtained by another research group and was not analyzed simultaneously in this study. Therefore, we should prepare our own samples for future studies. Third, the subjects in this study have had diabetes for many years and may not have autoreactive T-cells in the peripheral blood collected. On the other hand, Tregs have been reported to suppress GAD-responsive T-cells in patients with type 1 diabetes who have had the disease for more than 5 years [49]. Therefore, it has been reported that Tregs suppress GAD-reactive T-cells in patients with type 1 diabetes mellitus more than 5 years after onset, and it is possible that some immune abnormalities may still be present in PBMCs over time. Finally, we randomly selected patients who visited an outpatient clinic within a limited time. Therefore, we did not match background factors such as age, gender, duration of disease, and diabetes type, which are limitations of this study.

## 5. Conclusions

In conclusion, in this study, we used the latest technology, BD Rhapsody, to analyze the pairing of α and β chains that constitute the TCR of PBMCs from patients with type 1 diabetes at the single-cell level. In this study, we identified genes that are upregulated in T-cells as well as TRB repairs. scRNA-seq has greatly improved our understanding of heterogeneity in various biological processes and has led to significant breakthroughs in the fields of immunology, oncology, and developmental biology.

## Figures and Tables

**Figure 1 cells-11-01623-f001:**
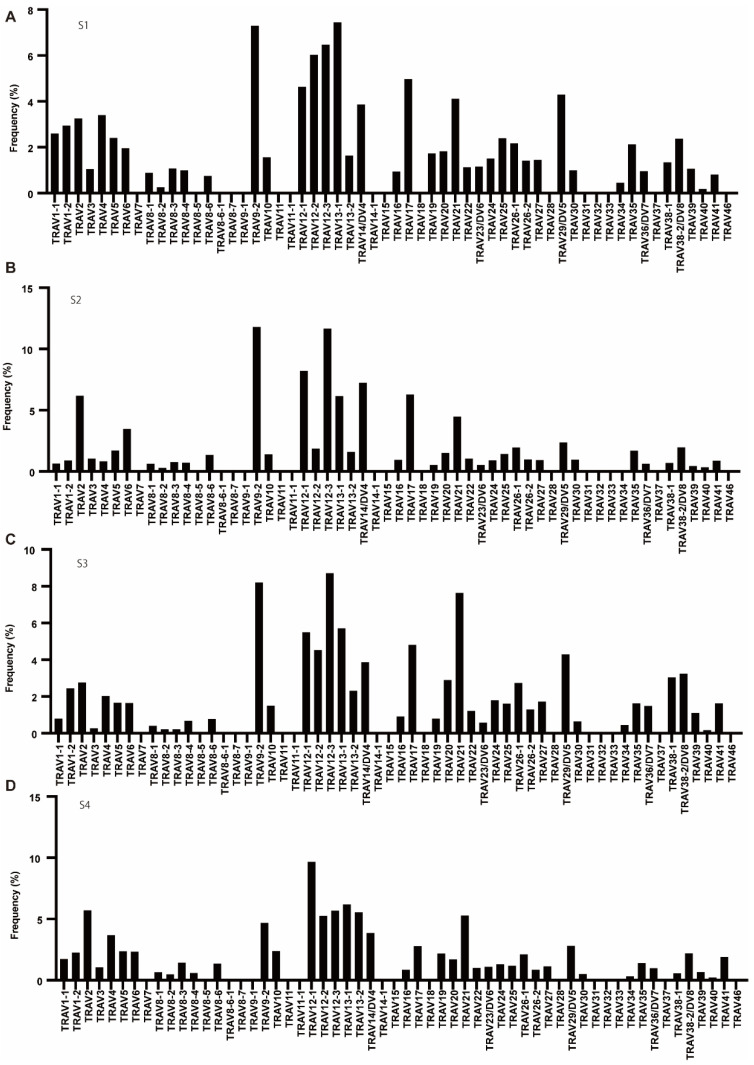
The frequency of TRAV. (**A**–**D**) Usage of TRAV in patients with T1DM (S1-S4). The numbers of TCR sequences bearing the respective TRAVs were counted. The percentage frequencies of TRAV were calculated and are shown.

**Figure 2 cells-11-01623-f002:**
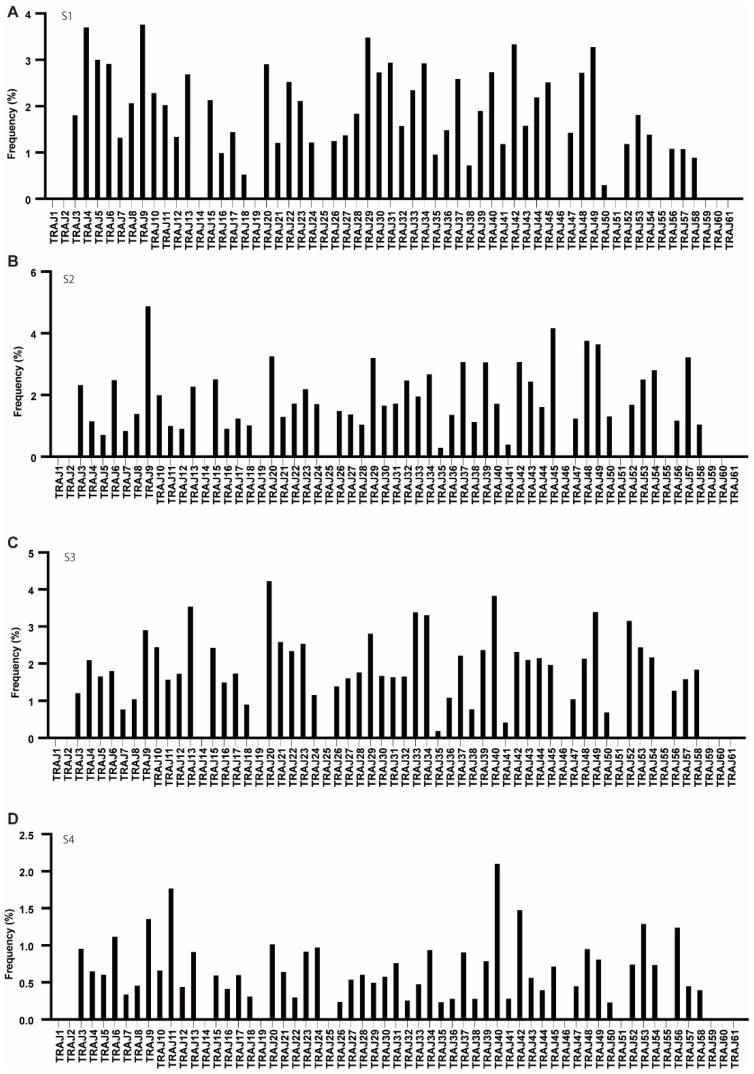
The frequency of TRAJ. (**A**–**D**) Usage of TRAV in patients with T1DM (S1-S4). The number of TCR sequences bearing the respective TRAJ was counted. The percentage frequencies of TRAJ were calculated and are shown.

**Figure 3 cells-11-01623-f003:**
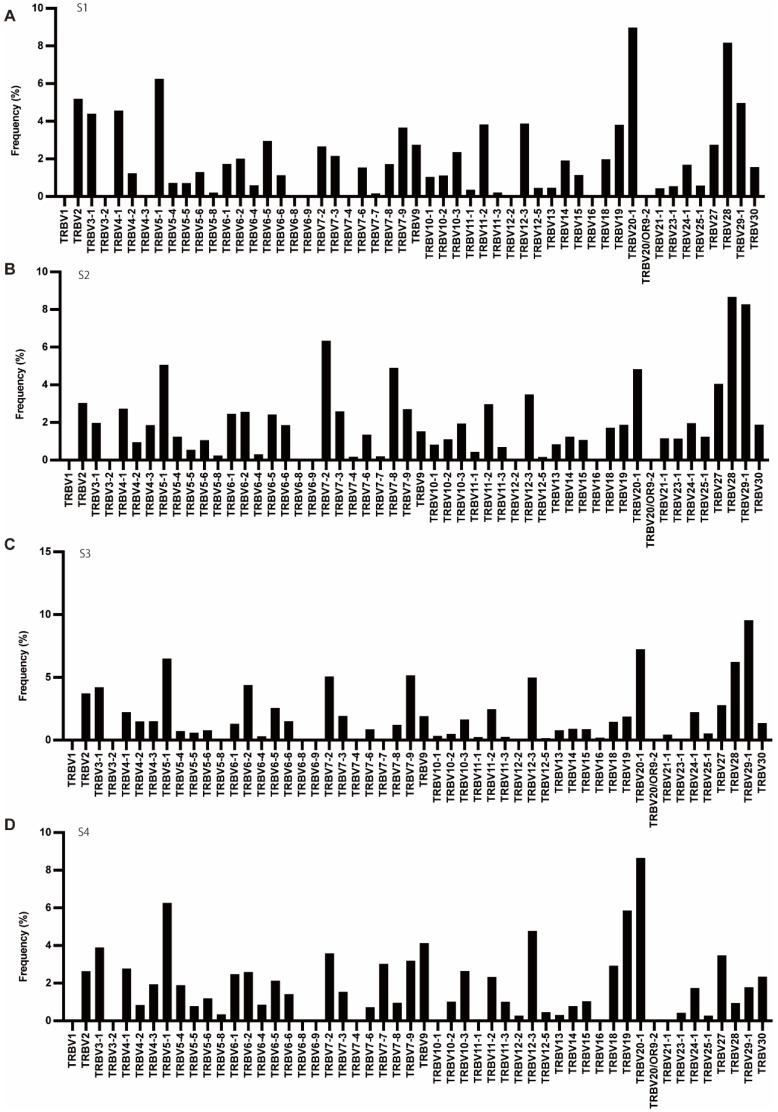
The frequency of TRBV. (**A**–**D**) Usage of TRBV in patients with T1DM (S1-S4). The number of TCR sequences bearing respective TRBVs was counted. The percentage frequencies of TRBV were calculated and are shown.

**Figure 4 cells-11-01623-f004:**
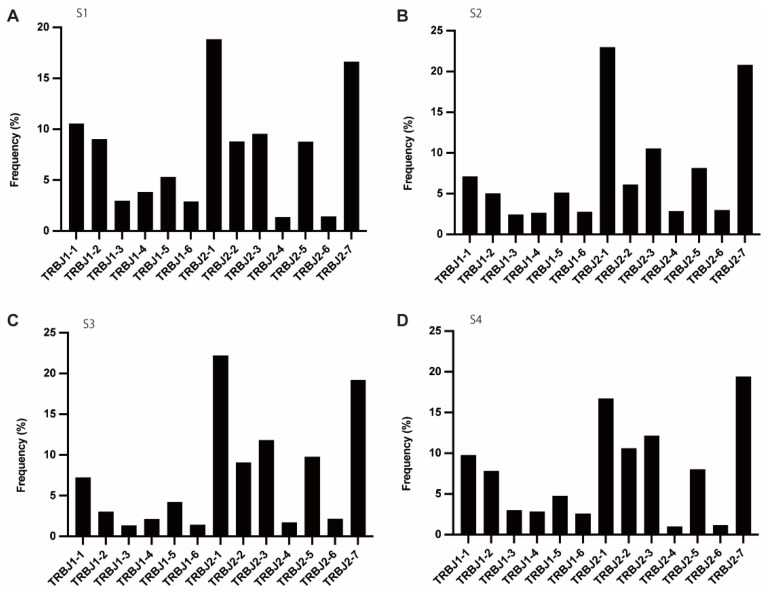
The frequency of TRBJ. (**A**–**D**) Usage of TRBJ in patients with T1DM (S1-S4). The number of TCR sequences bearing respective TRBJs was counted. The percentage frequencies of TRBJ were calculated and are shown.

**Figure 5 cells-11-01623-f005:**
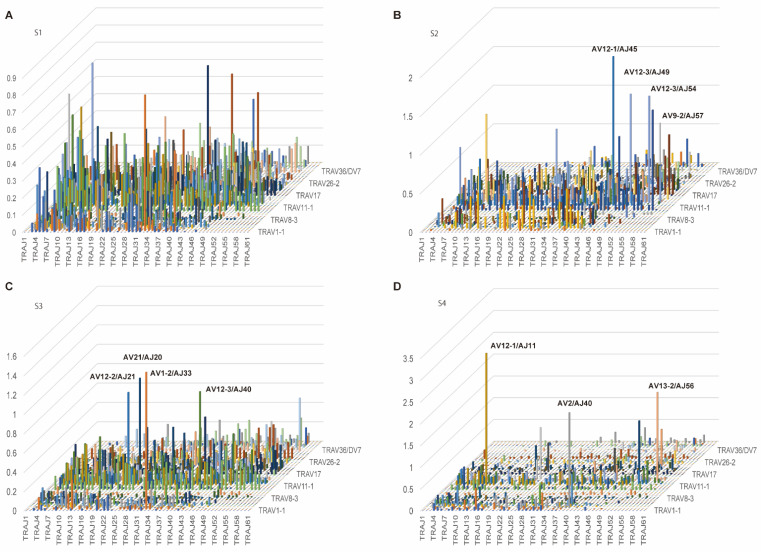
3D images of TRA repertoires. (**A**–**D**) The number of TCR sequence reads bearing a given gene recombination of TRAV with TRAJ was counted. The mean percentage frequencies are shown in the 3D bar graph. The X-axis and Y-axis indicate the TRAJ and TRAV, respectively.

**Figure 6 cells-11-01623-f006:**
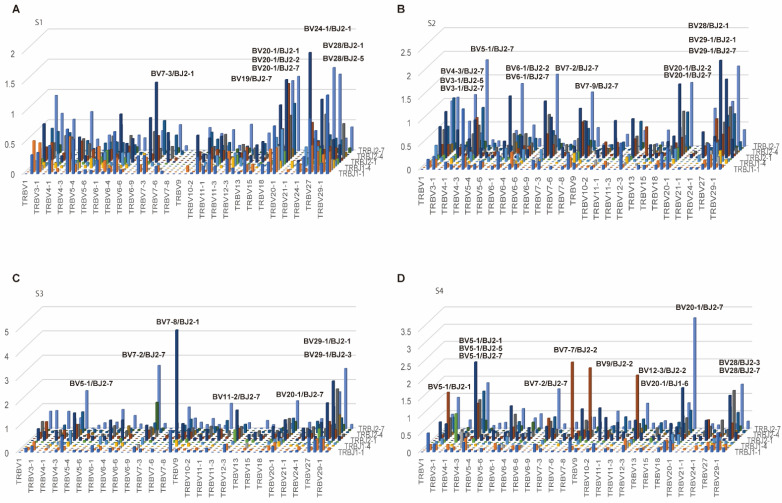
3D images of TRB repertoires**.** (**A**–**D**) The number of TCR sequence reads bearing a given gene recombination of TRBV with TRBJ was counted. The mean percentage frequencies are shown in the 3D bar graph. The X-axis and Y-axis indicate the TRBV and TRBJ, respectively. The names of high repertoires of more than 1% are listed.

**Figure 7 cells-11-01623-f007:**
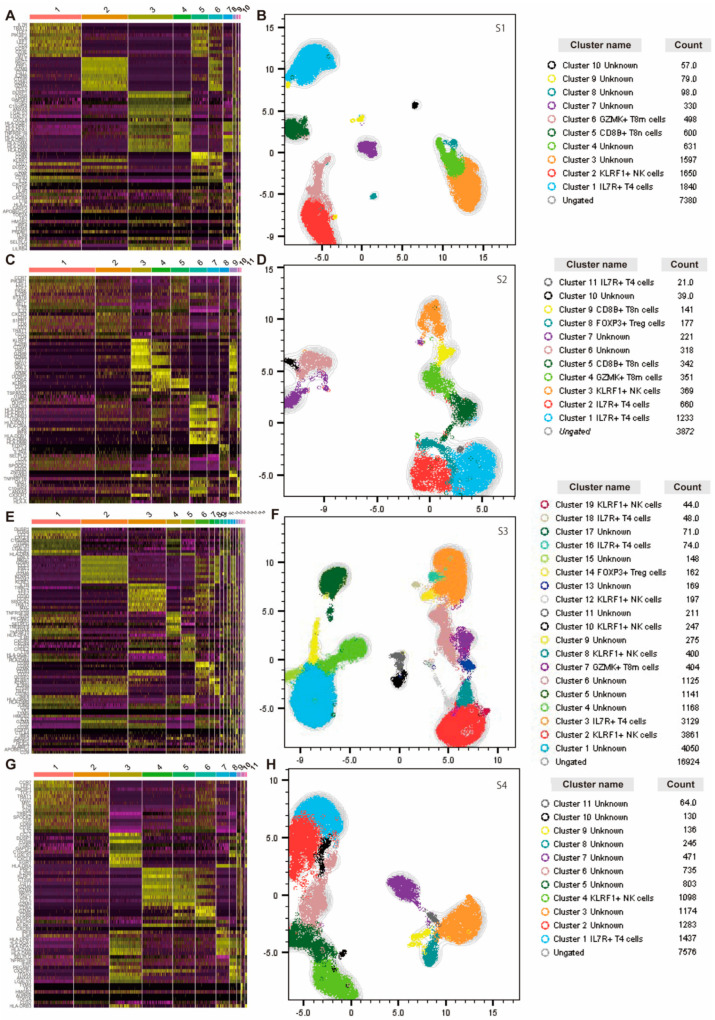
Seurat analyses of PBMCs. The Seurat plug-in in SeqGeq was used to cluster all PBMC data from patients with T1DM and project the clusters onto UMAP. A heatmap displaying expressed genes within the identified clusters, UMAP, and clusters is shown. (**A**,**B**) Heatmap, UMAP, and cluster name of S1 are shown. (**C**,**D**) Heatmap, UMAP, and cluster name of S2 are shown. (**E**,**F**) Heatmap, UMAP, and cluster name of S3 are shown. (**G**,**H**) Heatmap, UMAP, and cluster name S4 are shown.

**Figure 8 cells-11-01623-f008:**
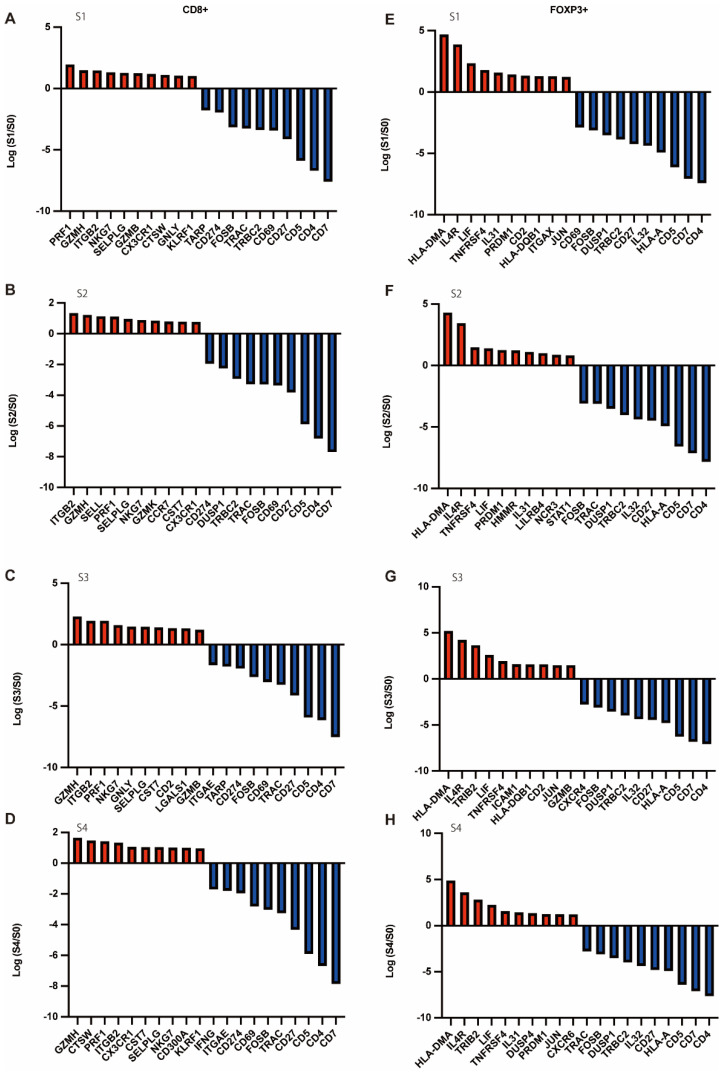
Upregulated and downregulated gene expression in CD8^+^ T cells and FOXP3^+^ T cells. The top 10 upregulated and downregulated genes of CD8^+^ T cells and FOXP3^+^ T cells in patients with T1DM compared to those in healthy subjects are shown. Gene expression was generated on a logarithmic scale. (**A**–**D**) Top 10 upregulated and downregulated genes of CD8^+^ T cells in S1-4. (**E**–**H**) Top 10 upregulated and downregulated genes of FOXP3^+^ T cells in S1-S4.

**Table 1 cells-11-01623-t001:** Clinical characteristics of the study patients.

Sample	1	2	3	4
Sex	Male	Female	Male	Male
Type	1A	1B	1B	SPIDDM
Age, yrs	71	68	47	49
Disease duration	16	46	19	7
Height, cm	169	150.2	181	168.4
Body weight, kg	57	45	100	76
Body mass index, kg/m^2^	20.0	19.9	30.5	26.8
Fasting plasma glucose, mmol/L	7.2	4.3	11.3	7.3
Hemoglobin A1c, %	8.1	7.5	7.5	8.5
C-peptide, mmol/L	<0.01	<0.01	<0.01	0.301
Creatinine, mmol/L	65.4	66.3	86.6	84.9
Estimated GFR, mL/min/1.73 m^2^	79.4	58.5	66.1	66.8
Urine albumin to creatinine ratio, mg/gCr	3673	64	8	19
Anti-GAD antibody	237	<5	<5	18.1

Cr, creatinine; GAD, glutamic acid decarboxylase; GFR, glomerular filtration rate; SPIDDM, slowly progressive insulin-dependent diabetes mellitus.

**Table 2 cells-11-01623-t002:** Shannon-index H′.

	TRA	TRB
S1	10.80	10.83
S2	11.62	11.69
S3	10.26	10.57
S4	11.37	11.25
S1 FOXP3^+^	5.37	5.57
S2 FOXP3^+^	6.27	6.35
S3 FOXP3^+^	5.62	6.14
S4 FOXP3^+^	5.84	5.77
S1 CD8^+^	9.17	9.14
S2 CD8^+^	9.74	9.77
S3 CD8^+^	8.24	8.61
S4 CD8^+^	9.44	9.37

TRA, T-cell receptor alpha; TRAB, T-cell receptor beta.

**Table 3 cells-11-01623-t003:** TCR clonotype of CD8^+^ T cells.

	Clone ID	Frequency (%)	TRA			TRB		
			TRAV	CDR3	TRAJ	TRBV	CDR3	TRBJ
S1	1-1	7.9	V27	AGAISNNDMR	J43	V9	ASSVVGSGTDEQF	J2-1
	1-2	7.4	V13-1	AASGSSASKII	J3	V6-5	ASSYSGQGSYT	J1-2
	1-3	4.0	V17	ATDSGGYQKVT	J13	V19	ASRLTGAGANVLT	J2-6
	1-4	3.6	V1-1	AVRDLDGGFKTI	J9	V10-3	AISEPEGNTEAF	J1-1
	1-5	3.4	V14/DV4	AMRRPSGGYNKLI	J4	V19	ASNAGYNEQF	J2-1
	1-6	3.1	V13-1	AASWDNAGNMLT	J39	V12-3	ASSDGTGGYEQY	J2-7
	1-7	2.7	V12-2	AVNPRRGFKTI	J9	V27	ASSLGLAGGYEQF	J2-1
	1-8	2.6	V6	ARASYGGATNKLI	J32	V9	ASSVTFERVPGANVLT	J2-6
	1-9	2.6	V1-1	APDTGRRALT	J5	V20-1	SARVVTGSSYEQY	J2-7
	1-10	2.6	V17	ATDMEEGGSQGNLI	J42	V19	ASNAGYNEQF	J2-1
S2	2-1	18.1	V12-1	VVRARPPLPWSGGGADGLT	J45	V7-2	ASTPPSSPGYEQY	J2-7
	2-2	10.3	V12-3	VPGGSASKII	J3	V20-1	SARGRPAGEQF	J2-1
	2-3	9.1	V6	ALKGYSGGYQKVT	J13	V28	ASSFSDRVNQPQH	J1-5
	2-4	8.1	V17	ATEGDSNYQLI	J33	V7-3	ASSSGTGDSLH	J1-6
	2-5	5.6	V12-3	AMSDYGGATNKLI	J32	V5-1	ASSPGRDRGSYEQY	J2-7
	2-6	5.2	V21	AVSPLSSGSARQLT	J22	V7-2	ASSLVSGPTYEQY	J2-7
	2-7	4.8	V9-2	AFDGGGATNKLI	J32	V4-2	ASSPGLGQPQH	J1-5
	2-8	4.4	V5	AESSGTGKLI	J37	V24-1	ATSDPAGGRADTQY	J2-3
	2-9	4.3	V12-1	VVNPRGSTLGRLY	J18	V10-2	ASSAGQGEAF	J1-1
	2-10	3.5	V14/DV4	AMQIDSWGKLQ	J24	V29-1	SVEDPHMDTQY	J2-3
S3	3-1	5.6	V38-1	AFSGGYQKVT	J13	V7-9	ASSLAGEGSGTGELF	J2-2
	3-2	3.4	V2	AVEDLLNSGYSTLT	J11	V6-2	ASSLRDSSYEQY	J2-7
	3-3	3.3	V21	AQGAYKLS	J20	V7-6	ASSPREAYEQY	J2-7
	3-4	2.7	V14/DV4	AMREGGSGYSTLT	J11	V2	ASSDRRGSSTDTQY	J2-3
	3-5	2.5	V27	GLN	J41	V20-1	SALRSGELF	J2-2
	3-6	2.4	V12-3	AMSGNQFY	J49	V28	ASRRFTGTDTQY	J2-3
	3-7	2.3	V12-3	AMTAGTYKYI	J40	V29-1	SADSSVGFHNEQF	J2-1
	3-8	2.3	V14/DV4	AMREYGNQFY	J49	V5-4	ASSRGQQPSYEQY	J2-7
	3-9	2.2	V12-2	AVNNQAGTALI	J15	V4-3	ASSQDLGANTEAF	J1-1
	3-10	2.1	V38-2/DV8	AYRSRGDMR	J43	V27	ASSFLAGATGELF	J2-2
S4	4-1	9.3	V10	VVSAFFSGGSYIPT	J6	V5-1	ASSSSRDRGNYEQY	J2-7
	4-2	5.5	V21	AVKGGSEKLV	J57	V7-8	ASSLVGLESYNEQF	J2-1
	4-3	3.7	V12-1	AVNLNTGFQKLV	J8	V2	ASRGYSYEQY	J2-7
	4-4	3.1	V12-3	AMVRAGGYNKLI	J4	V6-6	ASRSERESPISNEQF	J2-1
	4-5	3.1	V5	AALSGGSYIPT	J6	V4-3	ASSQGLREGLGEQY	J2-7
	4-6	3.1	V14/DV4	AMRNKSWGKFQ	J24	V3-1	ASSQEIVRTSGENTGELF	J2-2
	4-7	3.0	V6	ALGHSSASKII	J3	V20-1	SARDRDSSSYEQY	J2-7
	4-8	2.9	V21	AVASNFGNEKLT	J48	V29-1	SVAAGAQTQY	J2-5
	4-9	2.3	V2	AVEERIMGTYKYI	J40	V20-1	SARGVAANPYEQY	J2-7
	4-10	2.3	V12-1	VVPYNTDKLI	J34	V5-6	ASKPPGGSIYEQY	J2-7

CDR3, complementarity-determining region 3; TRA, T-cell receptor alpha; TRAJ, TRA joining; TRAV, TRA variable; TRB, T-cell receptor beta; TRBJ, TRB variable; TRBV, TRB variable.

**Table 4 cells-11-01623-t004:** TCR clonotype of FOXP3^+^ T cells.

	Clone ID	Frequency (%)	TRA			TRB		
			TRAV	CDR3	TRAJ	TRBV	CDR3	TRBJ
S1	1-1	6.7	V12-3	AMRFKSGYNKLI	J4	V18	ASSPPTSGASYEQY	J2-7
	1-2	6.3	V12-2	AVNIRDSSYKLI	J12	V20-1	SARSRLAVSGELF	J2-2
	1-3	6.0	V12-3	AMSDSGGGADGLT	J45	V3-1	ASSQRGGTQY	J2-3
	1-4	5.9	V12-1	VGLTNAGKST	J27	V11-2	ASSLGTQTTNEKLF	J1-4
	1-5	5.6	V2	AVEGGSGNTGKLI	J37	V2	ASSEEGNTEAF	J1-1
	1-6	4.6	V9-2	ATTRYSGAGSYQLT	J28	V28	ASTGTTSINEQY	J2-7
	1-7	4.1	V16	ARNFGNEKLT	J48	V12-3	ASSSRGGDNQPQH	J1-5
	1-8	3.2	V25	GRSGSARQLT	J22	V30	AWNRQGANTGELF	J2-2
	1-9	3.1	V13-1	AAPTIGRSKLT	J56	V7-3	ASSPLSSGANVLT	J2-6
	1-10	3.0	V4	LVAFDTGRRALT	J5	V23-1	ASSPPKFELLRAV	J2-7
S2	2-1	17.2	V9-2	ALSSNDYKLS	J20	V12-3	ASTLDGPGSPLH	J1-6
	2-2	9.3	V9-2	ALSGRNTGGFKTI	J9	V2	ASSRTKTDTQY	J2-3
	2-3	7.2	V35	AGPYSGAGSYQLT	J28	V28	ASSPSSGRASYEQY	J2-7
	2-4	5.4	V41	AVNAGNMLT	J39	V7-9	ASSSLDRGNIQY	J2-4
	2-5	4.5	V13-1	AASRPQGRRC*RTH	J45	V7-9	ASRLDATNEKLF	J1-4
	2-6	4.3	V38-2/DV8	AYRSYGAGNMLT	J39	V28	ASSQQGRQETQY	J2-5
	2-7	3.5	V12-1	VVRLNTGGFKTI	J9	V20-1	SARVGSTEKLF	J1-4
	2-8	2.9	V41	AVSSTPARQLT	J22	V6-6	ASSYSGSGSRRWHEQY	J2-7
	2-9	2.8	V38-2/DV8	APLGAGSYQLT	J28	V20-1	SASLMAVSYEQY	J2-7
	2-10	2.5	V12-1	VVNKQTGANNLF	J36	V28	ASRRRGGGTGELF	J2-2
S3	3-1	3.2	V21	GFSSGSARQLT	J22	V7-2	ASSFGRYEQY	J2-7
	3-2	2.6	V22	AANTPLV	J29	V12-3	ASSLLVDTQY	J2-3
	3-3	2.5	V21	AVTTGKST	J27	V20-1	SGQGTDTQY	J2-3
	3-4	2.1	V12-1	VVNMGGGFKTI	J9	V20-1	SASGGPGYNEQF	J2-1
	3-5	2.1	V13-1	AAGPMDSSYKLI	J12	V6-1	ASRLALTYNEQF	J2-1
	3-6	1.9	V13-1	AARGTSYGKLT	J52	V20-1	SARDPSSGLYNEQF	J2-1
	3-7	1.7	V21	AVRDDYKLS	J20	V20-1	SAGPGLAGVYEQF	J2-1
	3-8	1.6	V6	ALEDTGRRALT	J5	V25-1	ASTAPLGGLKQY	J2-3
	3-9	1.5	V21	AVYTSGSARQLT	J22	V6-5	ASSQGGGNTIY	J1-3
	3-10	1.5	V9-2	ALISSGSARQLT	J22	V10-2	ASSESRGSSNQPQH	J1-5
S4	4-1	4.4	V13-1	AAGRGNNRLA	J7	V12-3	ASSRTGGGYGYT	J1-2
	4-2	4.3	V10	VVRIAAISNTGKLI	J37	V24-1	ATSDHTQGRQGYT	J1-2
	4-3	3.8	V12-2	AVNGENFNKFY	J21	V12-3	ASSLAGTGVGYT	J1-2
	4-4	3.8	V2	AVEDRRQSGAGSYQLT	J28	V28	ASSFGFSNTEAF	J1-1
	4-5	3.8	V13-1	AASMNNQGGKLI	J23	V3-1	ASSQVRTGAYSNQPQH	J1-5
	4-6	3.7	V13-1	AASHGGSQGNLI	J42	V9	ASSVEVSGSYNEQF	J2-1
	4-7	3.4	V21	AGYNNDMR	J43	V4-1	ASSQGQGNYGYT	J1-2
	4-8	3.1	V1-1	ADRMDSNYQLI	J33	V20-1	SASPGQGADTQY	J2-3
	4-9	2.9	V12-2	AVRTKGGYQKVT	J13	V20-1	SPRGGGTEAF	J1-1
	4-10	2.8	V13-1	AASHGGSQGNLI	J42	V27	ASSYGVGGSIQY	J2-4

CDR3, complementarity-determining region 3; TRA, T-cell receptor alpha; TRAJ, TRA joining; TRAV, TRA variable; TRB, T-cell receptor beta; TRBJ, TRB variable; TRBV, TRB variable.

## Data Availability

The new high throughput sequencing (HTS) datasets in this study have been deposited in the GEO database (GSE197456, https://www.ncbi.nlm.nih.gov/geo/query/acc.cgi?acc=GSE197456, accessed on 30 April 2022). The data presented in this study are openly available in [FigShare] at [https://doi.org/10.6084/m9.figshare.19744570.v1].

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
