# Peer review of "Characterization of Peripheral Blood TCR in Patients with Type 1 Diabetes Mellitus by BD RhapsodyTM VDJ CDR3 Assay"

_cells, 2022, doi:10.3390/cells11101623_

Round 1

Reviewer 1 Report

In this manuscript, authors describe TCR usage in patients with T1D. I have no specific major comments, while I did wish authors would back-correlate the amino-acid sequences found with potential target peptides in these patients to see if there is an overlap in both 1B patients, for instance. Please find my minor comments below. After incorporation, this manuscript is suitable for publication.

Minor comments

General

  • Please remove “Figure X” from each figure.
  • For each figure, add S1-S4 to each panel where applicable as done in Figures 1 and 2 for consistency.
  • Please deposit the generated raw sequencing data to a repository.

Specific

Line 84 – please specify type(s) of anticoagulants for blood draw.

Starting at line 90 – according to manufacturer’s instructions? If yes, please add. Also, this was performed on fresh blood, not cryopreserved cells?

Line 191 – please describe the formula/methodology used to determine H-index for each sample in the materials and methods section, or say it was part of data package X that is already described there.

Line 197 – please define significantly high, or the cut off. Judging from the graphs, I assume it’s at 1% (which is a decent cut-off), please add this information here. For figure 5 itself, perhaps it is useful to add the percentage of the high repertoires in the name, i.e. panel B would then have AV12-1/AJ45 (1.44%) or something. Please also incorporate these comments for the text describing figure 6 and figure 6 itself.

Line 215 – decapitalize THE

Line 235 – this figure is of too low quality, please improve for legibility.

Line 298 – the downregulation of CD27 and CD5 argues that the majority of cells analysed is of an effector (memory) cell phenotype and should be mentioned in the discussion.

Line 320 – capitalize TNFRSF4 as refs 33/34 are human studies.

Line 352 – please remove this last sentence as that is the premiss of most scientific work and does not anything as is, unless authors state how their results will aid in future research specifically.

Reviewer 2 Report

In this manuscript, Okamura and colleagues analyzed the TCR repertoire of four patients with Japanese Type 1 Diabetes Mellitus by single-cell VDJ sequencing. They described the major findings as follows: (i) The expression levels of the TRAV17 and TRAV21 in T1DM patients were higher than those in healthy Japanese subjects. (ii) Shannon index of CD8+ cells and FOXP3+ cells in T1DM patients was lower, and (iii) identification of gene expression of CD8+ and FOXP3+ cells in patients with T1DM in comparison to healthy subjects.

There is growing evidence that the abnormalities of T cell repertoires are involved in the development and progression of both autoimmune and allergic diseases. The focus of this work is important and represents a high translational aspect. To shed more light on this issue the authors investigated the TCR sequences of characteristic autoreactive T cells in Japanese patients with T1DM by single-cell VDJ sequencing. The study is of certain interest for the readers and the potential translational aspect is important.

However, the authors should address the following points:

Major:

  1. The important aspect is the medical anamnesis of the patients included in this study. The information was requested from the selected patients, and how were they selected? Are there any known pre-existing conditions? Is it known if they have had other diseases during the study period? This information would be important as other infections could also affect the T-cell repertoire. In addition, the number of analyzed patients is low (4 patients) and divided into three types of TIDM, so the heterogeneity of the results makes difficult to extrapolate conclusions.
  2. All results in Figure 1 to 5 only show the usage of TRAV, TRAJ, TRBV and TRBJ derived from individual patient with different types of TIDM. Is there sharing V or J segments among four patients with different types of TIDM, especially TRA and TRB VJ rearrangement pattern? Moreover, is there sharing CDR3 motif between CD8+ and Foxp3+ cells?
  3. In Fig. 7 the authors described gene expression of different cell clusters in four TIDM patients. It would be interesting to investigate whether these unknown cell clusters could be identified from different TIDM patients. However, the figure is so unsharp, and what do these cell clusters represent? In addition, what do the FOXP3+ cell represent?
  4. Lack of HLA evaluation and consequent study for establishing the relation between these HLA-typings and the TCR repertoire. Even if these experiments are beyond the scope of this manuscript, the authors could at least discuss this aspect.
  5. Discussion should be suggested a hypothesis why there are sequences in TIDM patients present in healthy control group.

Minor:

  1. Letter size of figure 6 and 7 is in most of the elements very small (unreadable). And Figure 7 has low quality.
  2. The CD8A positive and CD8+, and FOXP3 positive and FOXP+ should be consistent throughout the manuscript.

Round 2

Reviewer 2 Report

The manuscript has been further improved by the authors, and this is appreciated. A few concerns/needs remain:

  1. The Modified Figures have been added to supplementary  but not to the manuscript. These sharp figures should be clear in the manuscript.
  2. Q5 should be responsed. Similar to the data analysis point, my intent was that whether there are the same sequences presented in TIDM patients comapred with healthy control group. Please provide some description in Results or Discussion section.

Author Response

1. The Modified Figures have been added to supplementary but not to the manuscript. These sharp figures should be clear in the manuscript.   Response Thank you for your valuable comment. The modified figure dose was too large and did not upload properly. Please check the uploaded file again after correction.     2. Q5 should be responded to. Similar to the data analysis point, my intent was that whether there are the same sequences presented in TIDM patients compared with the healthy control group. Please provide some description in the Results or Discussion section.   Response Thank you for your valuable comment. I checked again and was not able to respond to Q5. According to your comment, we have examined TRA and TRB VJ rearrangement pattern of all cells between the healthy control and T1DM patients. However, similar to what is described in Q2 of 1st revision, there was no common pattern. Therefore, we have added the sentences described as below.   Results Taken together, there was no common pattern among each of the four samples and healthy controls.